# Role of Knee and Ankle Extensors’ Muscle-Tendon Properties in Dynamic Balance Recovery from a Simulated Slip

**DOI:** 10.3390/s22093483

**Published:** 2022-05-03

**Authors:** Héloïse Debelle, Constantinos N. Maganaris, Thomas D. O’Brien

**Affiliations:** Research Institute for Sport and Exercise Sciences, Liverpool John Moores University, Liverpool L3 3AF, UK; c.maganaris@ljmu.ac.uk (C.N.M.); t.d.obrien@ljmu.ac.uk (T.D.O.)

**Keywords:** muscle strength, rate of moment development, tendon stiffness, knee extensors, plantarflexors, fall prevention, balance, slip, gait perturbation, age

## Abstract

Participants exposed to a simulated slip with forward loss of balance (FLB) develop large lower limb joint moments which may be a limiting factor for those whose muscle-tendon units’ (MTUs) properties are deteriorated. Whether the age-related decline in these properties limits participants’ capacity to recover their balance following a slip with FLB remains unclear. We combined isokinetic dynamometry, ultrasound and EMG to understand how knee extensor and ankle plantarflexor muscle strength and power, rate of moment development, electromechanical delay, and tendon stiffness affected the balance of young (25.3 ± 3.9 years) and older adults (62.8 ± 7.1 years) when recovering from a single slip with FLB triggered whilst walking on a split-belt instrumented treadmill. Except for the patellar tendon’s stiffness, knee extensor and ankle plantarflexor electromechanical delays, older adults’ MTUs properties were deteriorated compared to those of young participants (*p* < 0.05). We found no significant relationship between age or the MTUs properties of participants and balance recovery. These findings provide additional support that neither maximal nor explosive strength training are likely to be successful in preventing a fall for healthy older adults, and that other type of interventions, such as task-specific training that has already proved efficacious in reducing the risk of falling, should be developed.

## 1. Introduction

Falls cause a serious risk of injuries for older adults. Approximately a third of adults aged over 65 and half of those aged 80 and over fall at least once a year, meaning that in the UK alone over 4 million falls happen every year in the elderly. In 2020–2021, about 5% of these falls (n = 216,075) were severe enough to require emergency hospital admission [1], and can have serious health consequences such as long-term functional and cognitive decline, including fear of falling, task avoidance, loss of autonomy [2,3] and injuries including fractures [4] or can be lethal [4,5]. Together, these have an important financial impact on health services [6]. With the ageing population, fall prevention has become a topic of growing importance in the last twenty years. Although research has been conducted to prevent the initial loss of balance that leads to a fall by understanding and reducing the environmental (e.g., home hazards) and/or intrinsic risk factors for falls (physical capability, vision) [7], it is not always possible. When the balance is disturbed, fall avoidance relies on the person’s ability to recover their balance. Therefore, understanding the physical requirements of balance recovery following a gait perturbation may allow designing specifically targeted and thus more effective fall prevention interventions.

Following a loss of balance, it is necessary to quickly arrest the momentum of the body and maintain the centre of mass within the boundaries of the base of support. Studies investigating the biomechanics of gait following induced perturbations showed that the adjustments made are different for each type of perturbation. Specifically, participants recovering from a trip developed, depending on the timing of the perturbation, larger hip extensor, knee flexor or extensor, and ankle plantarflexor (PF) moments than in normal gait [8,9]. When exposed to a slip with backward loss of balance (BLB), participants tend to develop larger than normal knee and hip extensor moments, while inconsistent changes in ankle moments have been reported [10,11,12,13]. We recently showed a more complex profile of joint moments during recovery from a slip with forward loss of balance (FLB) [14]. On average, participants displayed significantly larger knee extensor (KE) and lower PF moments than normal, but a correlation analysis showed that those participants who developed lower than the mean (closer to normal) KE and larger than the mean PF moments were more stable on the following step. Additional results show that particularly at the knee joint, the simulated slip led to larger joint angular velocities accompanying the equal to or larger than normal joint moments in the step following the perturbation. This is indicative of greater joint power development from the knee extensors during that stance phase (see Appendix A). This pattern of larger KE and PF moments and power persisted until the second step following the slip. Recovery strategies were improved and strength demands reduced with repeated practice [14], but following the first exposure to a balance perturbation, participants will likely require adequate muscle-tendon unit (MTU) properties to meet the joint moment and power demands of a successful recovery strategy. Additionally, following the last slip, older adults produced significantly lower PF moments than young adults, with lower joint moments associated with poorer balance. Therefore, a decline in the force generation capacities of older adults may hinder their balance recovery following a slip with FLB.

It is well accepted that the ability to perform many important functional daily activities declines with ageing as a direct consequence of the deterioration in MTUs properties and function [15,16]. This includes the ability to maintain postural balance when performing unstable tasks (single leg and tandem stance), which is associated with the strength of the plantarflexor muscles and with the mechanical properties (stiffness) of their in-series tendon [17]. In dynamic situations, older adults’ weaker knee extensor and plantarflexor muscles and more compliant quadriceps femoris tendon contributed to their reduced balance recovery from a lean-and-release task compared to younger adults [18]. Studies investigating the MTUs requirements of balance recovery following a gait perturbation in older adults have demonstrated that participants with weaker knee flexors [19] KEs [19,20] and PFs [20,21], and those with lower AT stiffness [21] were at higher risks of falling when exposed to a trip or to a slip with BLB.

Recovery from a loss of balance may not only be associated with muscle strength but also with the rate of joint moment development [22], power [23] and electromechanical delay (EMD) [24] that all decline with ageing and, particularly for the lower limbs muscles, have been linked with increased risks of falling in elderly [20,25,26]. It has been shown that for elderly participants muscle power better predicts the outcome of a slip with backward loss of balance (fall or recovery) than muscle strength [27] which is not surprising as gait perturbations trigger very fast muscle response [8,28], and muscle power deteriorates at a faster rate than strength with ageing [23]. Therefore, understanding whether it is the absolute strength (peak moment), power or the rate at which it is generated (RMD) or transferred (EMD) that primarily determines the outcome of a perturbation is vital for the development of exercise interventions that should specifically target the required characteristics of muscle and tendon function.

A recent literature review on the mechanical properties of the leg extensor MTUs in older participants showed an overall agreement towards a reduced stiffness of both the patellar and Achilles tendons (PT and AT, respectively) stiffness [29], that also contributes to the functional decline of older adults. First, more compliant tendons lead to increased shortening and shortening velocities of the sarcomeres during contractions, thus affecting the maximum joint moment production and rate of moment development capacity as per the force-length-velocity relationships [30,31,32]. Second, tendon stiffness is also associated with joint power [33] and EMD [34], therefore compliant tendons that delay and slow the transmission of the mechanical force, can contribute to reducing the contraction velocity of older adults. Hence, the mechanical properties of elderlies’ tendons, their role in the functional decline and thereby in the increased risk of falling of this population, also need to be investigated to design effective fall prevention interventions.

Some previous research has supported the assumptions that the age-related deterioration of MTUs properties limits older participants’ capacity to successfully recover their balance following trips [22,35] and slips with BLB [19,27]. However, contrasting findings exist. Specifically, it has been shown that MTUs properties were not significantly different between stable and unstable participants during a lean-and-release task [36], that the maximum knee flexors and extensors strength was not different between fallers and non-fallers when exposed to slips with BLB [27], and that both strong and weak older adults were at risk of falling following a trip although through different mechanisms [35]. The links between MTUs properties and balance recovery following slips with FLB have not been established yet. Given the complex profile and influence of joint moment and power development on recovery from the first exposure to a slip with FLB, establishing this understanding may be important for developing exercise interventions to decrease the risk of falling by specifically targeting these MTUs properties.

Therefore, the present study aimed to determine the relationship between the KE and PF MTUs properties and participants’ ability to remain stable following a single slip with FLB. We hypothesised that (1) older participants would have declined MTUs properties (strength, power, RMD, EMD and tendon stiffness) compared to young participants, and (2) that the PF and KE MTUs properties would be related to the balance of our participants following a single slip with FLB. Predominantly, as (a) our previous research found that the KE produced larger joint power and the PF produced larger joint moments, and (b) the RMD of PF and KE muscles has been shown to be lower in older fallers compared to young adults [22,37], we hypothesised that the knee joint power, isometric peak PF moment and RMD of both PF and KE muscles would be most strongly correlated with balance recovery.

## 2. Materials and Methods

### 2.1. Participants and Protocol

The study was conducted with 14 young (7 males, mean age 25.3 ± 3.9 SD years, height 177.2 ± 8.3 cm, body mass 72.8 ± 10.6 kg) and 14 older (2 males, age 62.8 ± 7.1 years, height 161.7 ± 7.6 cm, body mass 66.9 ± 12.1 kg) adults who were a sub-sample of a larger cohort from our previous study (17 young and 17 older participants) [14] that volunteered to have their MTUs properties tested during an additional testing session. Participants were excluded from the study if they were not able to walk unassisted for a minimum of 15 min (as this was the average duration of walking during the testing session), could not comfortably walk on an instrumented treadmill (M-Gait, Motekforce Link, Amsterdam, The Netherlands) at 1.2 m·s^−1^ during the familiarisation period (5 min), had suffered in the last 6 months from any musculoskeletal injury, had a musculoskeletal surgery in the last 2 years, or had any neurological or balance disorder.

To examine whether participants’ MTUs properties were linked to their balance following a slip with FLB, we quantified their knee extensors and ankle plantarflexors muscle strength, power, RMD, EMD and tendon stiffness, as well as balance recovery. Tests were performed across two testing sessions 8.1 ± 6.7 days apart, starting with the MTUs properties evaluation.

This study was carried out with the approval of the Liverpool John Moores University (18/SPS/017) and National Health Service (18/NW/0700) ethics committees. Written consent was obtained in accordance with the declaration of Helsinki.

### 2.2. Evaluation of Balance Recovery

The method used to evaluate participants’ balance following slips with FLB has previously been described elsewhere [14,38]. Briefly, after five minutes walking on a split-belt treadmill at 1.2 m·s^−1^ for familiarisation, participants were exposed to an unexpected slip-like perturbation, simulated by a sudden acceleration of the ipsilateral belt to the stance foot. Previously reported results showed no anticipatory changes in gait, and very good consistency for both the perturbation timing and timing of peak instability recorded [38]. Participants wore a full-body safety harness that was attached to a frame above the treadmill to prevent a dangerous fall from happening should they not recover their balance. We used a 6DoF marker set and the 3D coordinates of 68 retroreflective markers were recorded by 12 motion capture cameras (120 Hz; Vicon Motion Systems, Oxford, United Kingdom). Participants’ balance was measured as their margin of stability (MoS) [39] that represents the anterior-posterior distance between their extrapolated centre of mass and the anterior boundary of their base of support represented by the second toe marker. A positive MoS indicated that the participants were stable, a negative MoS that they were unstable.

To avoid using multiple MoS measures and therefore increasing the risk of type 1 error in our statistical analysis or having to considerably correct α and risk not detecting existing effects, in the present paper we report our participants’ balance as the MoS averaged over the first three recovery steps. This choice is based on several factors. First, we showed that following this slip with FLB, our participants’ MoS decreased until the third recovery step before it gradually returned to normal levels (average of 5 consecutive steps recorded in unperturbed walking conditions after a familiarisation period) on the seventh recovery step [14]. Additionally, joint moments were significantly different from normal from the perturbed step to the first recovery step and correlated to participants’ balance recovery. Larger joint moments were detected until the second recovery step. Therefore, if MTU properties are important to balance recovery following the slip it is likely to be from the first to the third recovery step.

### 2.3. Evaluation of MTUs Properties

#### 2.3.1. Set Up

Muscle strength, joint power, RMD, EMD and tendon stiffness were tested on participants’ right leg. Participants were given instructions, familiarisation trials and rest periods between the different contraction types and trials. To avoid inducing fatigue due to the repetition of familiarisation trials, sub maximal trials were conducted to ensure that participants were comfortably positioned on the dynamometer, that the set up was not limiting their force production capacity (no discomfort reported) and, for each contraction type, that they followed the instructions provided (ramped or maximum). Participants then performed one maximal contraction for familiarisation before we started recording the trials. A trial was repeated if it did not reach 75% of the maximum moment produced over 3 trials. A maximum of one additional repetition was done per contraction type. Rest period between conditions lasted 5 min and 1 min between similar trials. Previous research in elderly participants showed that 30 s rest periods were sufficient for between trial strength recovery [40].

Participant’s skin was thoroughly cleaned and shaved to collect EMG signals from the rectus femoris, biceps femoris, lateral gastrocnemius and tibialis anterior muscles using surface electrodes, following the SENIAM guidelines for electrode locations [41]. EMG signals were sampled at 1600 Hz simultaneously with joint moments and angular velocities in AcqKnowledge software (Acqknowledge, Biopac MP150, Biopac Systems, Goleta, CA, USA). Raw EMG signals were filtered with a zero-phase shift 6th order Butterworth bandpass filter with a cut-off frequency from 20 to 500 Hz, full wave rectified, and the linear envelop of the signal was extracted by applying a zero-phase shift 6th order Butterworth lowpass filter with a cut-off frequency of 30 Hz.

Knee extensors were tested with participants seating on an isokinetic dynamometer (IKD) (Humac Norm Computer Sports Medicine Inc., Stoughton, MA, USA) chair with their hips flexed at 85°. Ankle plantarflexors (PFs) were tested with participants laying prone on the dynamometer bed with their ipsilateral hip and knee extended (corresponding to 180° and 0°, respectively). The axis of rotation of the dynamometer arm was aligned with the joint’s axis of rotation during a submaximal contraction and the joint’s range of motion was passively determined beforehand. Net joint moments were corrected for the gravitational moment caused by the weight of the participant’s limb and dynamometer arm.

#### 2.3.2. Quantification of Muscle Strength

Knee extensors and ankle plantarflexors strength (KE_MAX_ and PF_MAX_, respectively) was measured as the maximum of three representative ramped isometric contractions performed with the joints at 90° and at optimum joint angle, typically reported for these joints for young and older participants (75° for the knee [42,43] and participant’s maximum dorsiflexion angle for the ankle [44,45]).

Inspection of knee and ankle angles curves of the slipped leg during the perturbed stance show that at time of peak joint moment participants tend to maintain their knee angle and decrease their ankle dorsiflexion angle (see Appendix A). Therefore, in the present paper to better approximate the muscles operating lengths during the perturbation, KE_MAX_ is reported at 75° flexion and PF_MAX_ at 90°.

#### 2.3.3. Quantification of RMD and EMD

Knee extensors and ankle plantarflexors’ RMD and EMD were measured during three isometric contractions in which participants were asked to produce maximal rapid contractions with their knee and ankle at 90°, respectively. RMD was measured as the slope of the joint moment-time curve between 20% and 80% of MVC and the maximum RMD over the three contractions was used for statistical analysis. The onset of EMG activation was set at 2 standard deviations above the average EMG activity recorded at rest over 250 ms at the beginning of each trial before the contraction. The onset of muscle contraction was defined in the same way using the gravity corrected joint moment signal. This threshold has previously been used in the literature [46,47,48] with acceptable coefficient of variation between trials (CV < 6.5%) [49]. EMD was then calculated as the time difference between the onsets of EMG activity and joint moment and the minimum of three contractions was used for statistical analysis.

#### 2.3.4. Quantification of Joint Power

Peak joint power was quantified as the product of the maximal joint moment and angular velocity (in rad/s) of three consecutive concentric contractions performed over 100% of joint’s range of motion, at 60°·s^−1^, 120°·s^−1^ and 240°·s^−1^ for the ankle, and at 60°·s^−1^, 120°·s^−1^, 240°·s^−1^ and 300°·s^−1^ for the knee. Maximum joint moment during isokinetic contractions was extracted from the iso-velocity phase.

Inspection of gait data showed that during the perturbation, concentric contraction of the knee extensors was occurring at a knee angular velocity on average below 50°/s (see Appendix A). Therefore, to better approximate the joint angular velocity during the perturbation, knee joint power is reported at 60°·s^−1^ in the present paper. Additionally, knee joint power is also reported at 120°·s^−1^ as it was previously shown to discriminate fallers from non-fallers [27]. Although participants’ ankle angular velocity at push-off of the perturbed stance phase, corresponding to a concentric contraction of the plantarflexor muscles, reached ~250°·s^−1^ (see Appendix A), during the isokinetic tests most older participants did not reach their maximal joint moment within the iso-velocity portion of the concentric contraction at 240°·s^−1^ (n = 7) or did not reach the target velocity at all (n = 6). Therefore, ankle joint power is reported at 120°·s^−1^ here.

#### 2.3.5. Quantification of Tendon Stiffness

To avoid biasing measurements of tendon stiffness due to hypothesised differences in maximum strength between groups, tendons’ stiffness is reported at a common force region for all participants. Therefore, tendon stiffness is reported over the strongest 30% of the weakest participant (older adult), which corresponds to tendon’s force of 479.5 N–685.0 N for the Achilles tendon, and 830.0–1185.5 N for the patellar tendon. For each participant, the coefficients of a second order polynomial function (fitted to the data using the least square methods) were extracted and used to calculate the elongation of the participant’s tendon at the corresponding 70 and 100% of the weakest participant’s tendon force. Stiffness was subsequently quantified as the gradient of the resulting force-elongation relationship.

The PT force was calculated from a ramped isometric KE contraction by dividing the knee extensor moment (M_KE_) (corrected for both the gravitational moment and the antagonist co-contraction) by the moment arm of the PT (MA_PT_).

Net moment was measured during three ramped isometric contractions performed with the knee at 90° of flexion. The moment due to the antagonist co-contraction was calculated assuming a linear relationship between the amplitude of the EMG signal and the joint moment produced [50]. Briefly, the moment-voluntary activation level of the antagonist was calculated at every 10% of a maximum knee flexion effort with the knee flexed at 90°. The linear equation of the obtained moment-EMG curve was extracted and used to calculate the antagonist moment produced during the ramped isometric KE contractions. M_KE_ was then computed as the sum of the measured net and calculated antagonist moments. The strongest trial was used for further analysis. M_KE_ was extracted at every 10% of peak MVC.

The MA_PT_ length was measured individually during contraction to account for the known changes. Video recordings (iPhone X, 30 Hz, Apple, Los Altos, CA, USA) and linear B-mode ultrasonography (Esaote Mylab70, Esaote SPA, Genoa, Italy) were recorded synchronously to estimate the location of the PT’s line of action relative to the joint centre. A 5 V external trigger was used to synchronise the ultrasound videos (15–50 Hz depending on ultrasound settings), camera videos (30 Hz) and Acqknowledge data (see Figure 1). Two external markers were placed on the ultrasound probe at a known 2D (anterior-posterior (X) and vertical (Y)) distance from the origin of the transducer array, and parallel to the transducer array. An additional marker was added on the lateral femoral condyle which was assumed to represent the lateral projection on the skin of the knee joint centre. The camera lens was aligned with the lateral condyle and the dynamometer axis of rotation. The ultrasound probe was aligned with the tendon’s line of action, secured to the skin with inextensible tape and lightly held in place by the researcher to counteract the effect of gravity on the probe and cable’s mass. Images were extracted from the ultrasound and camera videos at every 10% of M_KE,_ and manually analysed in ImageJ software (ImageJ version 1.51j8, National Institutes of Health, Bethesda, MD, USA). From the ultrasound images, we extracted the 2D coordinates of the PT’s proximal (patellar apex) and distal (tibial tuberosity) insertions. From the video images, we extracted the 2D coordinates of the ultrasound probe and joint centre. All data were exported to Matlab version R2021a, in which we used a custom-made script to quantify MA_PT_ as the perpendicular distance between the PT’s line of action and the knee axis of rotation. The PT’s force was then calculated as the ratio M_KE_ on MA_PT_.

The PT’s length was measured as the distance between its proximal and distal insertions at rest and at every 10% of M_KE_. Its elongation was then calculated as the difference between the lengths measured during contraction and at rest.

The AT force was measured with a similar protocol. The plantarflexor moment (M_PF_) corrected for both the gravitational moment and the moment due to the antagonist co-contraction and the AT elongation were measured together during 3 ramped isometric contractions with the ankle at 90°. The joint moment due to the antagonist co-contraction was measured assuming a linear relationship between the amplitude of the EMG signal and the joint moment produced during one maximal isometric dorsiflexion contraction. The AT elongation was measured with linear B-mode ultrasonography (Philips EPIQ7, Amsterdam, The Netherlands) as the proximal displacement of the myotendinous junction of the medial gastrocnemius during the isometric contractions. Ultrasound videos were collected at 15–50 Hz depending on ultrasound settings. Care was taken to minimise movement of the calcaneus, and thereby displacement of the distal tendon insertion, by securely strapping the foot in the footplate. A custom-made arch support was used to help maintaining heel contact with the footplate during the contractions. The ultrasound transducer was aligned with the tendon’s line of action, secured to the skin with a custom-made probe holder, Velcro and inextensible tape, and held in place by the researcher during contractions.

It was not possible to evaluate the AT moment arm (MA_AT_) and tendon elongation during the same measurement as the ultrasound probe would have been out of plane with the axis of rotation. Therefore, to measure MA_AT_ three additional isometric ramped contractions were performed with the ultrasound probe located at 50% of the tendon’s length (M_PF-MA_). The camera’s lens was aligned with the intermalleolar axis facing the medial malleolus which was assumed to represent the ankle joint centre. To clear the camera’s field of view, participants’ contralateral knee was flexed at 90° during these contractions. Again, an external 5 V trigger was used to synchronise the different systems. The plantarflexor moment measured at every 10% MVC of the M_PF_ contractions was averaged over the 3 contractions and MA_AT_ extracted at the corresponding joint moments produced during M_PF-MA_.

The 2D coordinates of the tendon’s most proximal and distal ends on the ultrasound images, the coordinates of the joint axis of rotation and probe markers obtained during M_PF-MA_ were manually digitized using ImageJ and exported to Matlab. MA_AT_ was then calculated as the perpendicular distance between the tendon’s line of action and the ankle axis of rotation. The AT’s force was calculated at every 10% of M_PF_ as the ratio of M_PF_ on MA_AT_.

A summary of the tests performed for the evaluation of the MTUs properties is available in Table 1.

### 2.4. Statistical Analysis

To test whether there was an effect of age on the MTUs properties (KE_MAX_, PF_MAX_, knee power (at 60 and 120°/s), ankle power (120°/s), PFs’ RMD, KEs’ RMD, PFs’ EMD, KEs’ EMD, PT and AT stiffness) and on participants’ balance, we used Mann-Whitney U tests. We did not perform α corrections as the effect of age on the MTUs properties of our participants were investigated for each individual variables [51]. An independent sample t-test showed that there was no between group difference in participants’ body mass. Therefore strength, power and RMD are presented as non-normalised values.

To test whether participants’ age and their MTUs properties were correlated to their balance, we used bivariate Kendall’s tau (τ) correlations with BCa 95%CI. Significance was accepted at *p* < 0.05.

## 3. Results

Differences between young and older adults were detected as being statistically significant for knee extensor and plantarflexor strength (U = 32.000, *p* = 0.006 and U = 26.000, *p* = 0.001, respectively), knee power at 60°·s^−1^ (U = 22.000, *p* < 0.001), and 120°·s^−1^ (U = 11.000, *p* < 0.001), ankle power (U = 9.000, *p* < 0.001), knee extensor RMD (U = 15.000, *p* < 0.001), plantarflexor RMD (U = 20.000, *p* < 0.001) and Achilles tendon stiffness (U = 42.000, *p* = 0.017). Overall young participants were stronger, developed larger joint power and had larger RMD than older participants (Table 2).

There was no significant effect of age on participants’ MoS following the slip (U = 104.000, *p* = 0.804) (Table 2).

We found no significant correlation between the age or any MTU properties and the MoS of our participants (*p* > 0.05) (all correlations are reported in Table 3, primary relationships to test the hypotheses are shown in scatter plots in Figure 2).

## 4. Discussion

The aim of this study was to establish whether MTUs properties are important for recovery of balance following a slip causing a forward loss of balance, and if so, to identify which properties are most important. We hypothesised that older adults’ MTUs properties would be deteriorated compared to those of younger adults, which was mostly confirmed. We further hypothesised that the properties of the plantarflexor and knee extensor muscles and tendons, would be correlated with balance following a single slip, this hypothesis was rejected.

In agreement with numerous studies, we found that muscle strength [17,52,53], power [23,54], RMD [32] and Achilles tendon stiffness [17,53] were lower in older than younger adults. Particularly, percentage differences between our age groups were similar to those reported in the literature for PF strength (mean difference was 38%, and ranged from 29 to 52% in the literature [17,18], KE strength (mean difference was 37%, and ranged from 15 to 35% in the literature [55,56] and AT stiffness (mean difference was 31%, and ranged from 17 to 39% in the literature [17,53]. Despite a similar trend, the effect of age on RMD was larger in the present study (mean difference 64% compared to 35% in the literature [32]. As our group of older participants included a majority of women, and because the rate of moment development is significantly lower in females than in males [57], sex differences in the rate of decline may explain these differences. In this study, we found that the PT stiffness was not statistically different between ages despite the group median being 20% lower in older than younger participants. The literature shows a tendency towards a decreased patellar tendon stiffness with ageing [29] but similarly to the present study, large inter-individual differences have been observed leading to non-significant age effects [58,59] possibly resulting from a type 2 error and would require larger sample size to be appropriately tested. As mechanical loading affects the properties of tendons [60], additional evaluation of participants’ physical activities (type and levels), and therefore mechanical demand, is needed to better compare the tendons’ stiffness between young and older participants. Finally, we found no age-related differences in participants’ EMD for both the knee extensors and plantarflexors which is surprising considering the significant (Achille’s tendon) and non-significant (patellar tendon: median difference 20%) differences observed on the tendon stiffness. Both chemical (synaptic transmission, propagation of the action potential and cross bridges formation) and mechanical (force transmission along the active part of the in-series elastic components, aponeurosis and tendon) processes affect the EMD [61]. The present study only evaluated the stiffness of the free tendons that accounts for about a third of the EMD [34], therefore age-related modifications affecting the other components of EMD may have affected these results. Also, other authors reported non-significant changes in EMD with ageing [62] and suggested that it may not be reliable and sensitive enough to age-related changes to be considered an appropriate tool to measure force transmission along the MTUs. Other considerations, such as the lack of control of the joint rotation during the contractions, although externally limited by a non-extendable strap and unlikely to differ meaningfully between groups, may have contributed to overestimate the observed tendon’s lengthening and thereby to modifying the participants’ tendon stiffness. Future studies should therefore measure the inevitable joint rotation associated with IKD testing and its effect on the tendon lengthening evaluation.

The main results of the balance recovery data for this group of participants have been previously reported, showing that there was no significant effect of age on participants’ recovery from a slip with a FLB [14]. The present study further shows no significant difference between age groups in balance calculated as the average of the MoS measured on the first three recovery steps. This is rather surprising considering that multiple studies [18,22,63,64,65,66] using various perturbation types all concluded that the balance declines or the rate of falls increases with older age. Several factors may explain this difference and have been discussed in detail elsewhere [14]. Briefly, our participants were on average younger than the participants recruited in the aforementioned studies, it is therefore likely that our older participants were too young to detect an effect of age on their balance. To be included in this study, our participants had to be able to walk unassisted for at least fifteen minutes, therefore our older participants may also have been too highly functioning to significantly differ from young adults. Lastly, the instability created by the protocol used in this study might have been not mechanically demanding enough, as no fall was triggered, to discriminate young and older participants.

Although some studies have previously established clear relationships between MTUs properties and balance, the main finding of the present study was that our participants’ KE and PF MTU properties did not explain their balance following a slip with FLB. In a previous study investigating the effects of MTUs properties on participants balance during a lean-and-release task simulating a forward fall [36], it was shown that older stable participants were not stronger, nor did they have stiffer tendons than older unstable participants, which further supports our findings that the MTUs properties of participants might not be a discriminant factor for balance recovery. Using that same lean-and-release task [18], the relationship between MTU properties (muscles strength and tendons stiffness) and MoS of young and older participants was found to increase with task demand (0.429 ≤ r ≤ 0.522 in the lowest demanding task (i.e., recovering from small lean angle ~20°), and 0.538 ≤ r ≤ 0.663 in the highest demanding task (i.e., recovering from a large lean angle ~26°)). The perturbation we imposed within gait was notably more complicated for the participants to respond to. Nonetheless, it is possible that the perturbation triggered in our study might not have been sufficiently challenging for participants to rely primarily on MTUs properties. Additionally, although the MTUs properties were related to participants’ balance during the lean-and-release task, they only contributed to 35 to 55% of the balance recovery [18], therefore other factors must contribute, at least in equal, if not higher proportions, to the participants’ ability to recover their balance. For the same highly demanding task (lean angle ~26°), it was also shown [36] that the balance at touch-down of the first recovery step was highly (r^2^ = 0.937) correlated with balance at the end of the downward phase of that same step, and that the participants who were classified as stable were those who successfully adjusted their horizontal GRF during push-off and landing to better control their centre of mass. Therefore, it might not be the MTUs properties that discriminate stable from unstable participants but rather their ability to enact adequate recovery strategies in reaction to a perturbation.

The finding of this study adds weight to questions over the value of exercise-based interventions to prevent fall occurrence. As previously discussed by other authors [35,67], although we understand well the mechanisms and direct effects of exercise interventions on health (and thereby musculoskeletal health), the mechanisms by which exercise interventions decrease the risks of falling, demonstrated with different type of interventions, are poorly understood, which hinders our ability to design effective interventions. Based on previously gathered evidence that the KE and PF joint moments were correlated (positively or negatively) with balance recovery following exposure to a single slip with FLB [14], and that participants quickly developed important knee concentric efforts following that slip (see Appendix A), we designed this study to relate the MTUs properties including joint moments, RMD and power to the kinetic demands observed in balance recovery. Despite this care and specificity in selection of predictor variables, we did not find any link between the MTUs properties of our participants and their balance, and thus we cannot provide new guidance to design specific exercise interventions aiming to decrease the risk of falling by specifically targeting these MTUs properties. On the contrary, our results support the idea that a task-specific intervention in which our participants would be exposed to multiple occurrences of gait perturbations would be more beneficial for balance recovery. Indeed, we previously showed that when our participants were exposed to 10 consecutive slips with FLB, they quickly refined their recovery strategy so that by the last slip they returned to a normal balance from the second recovery step without relying on larger than normal joint moments [14]. Although our study has not been designed to document the degree to which improvements in gait stability are retained long-term, our findings support the hypothesis that there might be a greater benefit in preventing locomotory falls by training protocols that challenge and train the participants ability to recover from a gait perturbation, rather than exercise interventions to improve MTUs properties.

Some limitations have been identified in this study. First, as identified above, our participants were probably too young and highly functioning to detect differences in balance recovery between our age groups. This is an important limitation as it may have affected our correlations between MTUs properties and balance recovery. Therefore, our results are only valid in the context of higher functioning older individuals and cannot be generalised to frail populations. However as the risk of falling increases with fall history [68], this population of relatively young and functioning older participants should also be targeted in fall prevention interventions to avoid a first fall. Also, we did not account for cofounding variables such as participants’ individual characteristics (sex or activity levels) that may have influenced our variables. Particularly our older participants were mainly females (n = 12), and it is well accepted that sex affects the muscles [69] and tendons [70] mechanical properties of participants, with females developing less strength and having more compliant tendons than male participants. Therefore, our between groups analysis may have been biased by the greater number of older female participants and future studies should focus on homogeneous samples. To run the testing session smoothly, we did not randomise the order of the muscle contractions, nor did we perform a final MVC to evaluate the effect of fatigue with repeated contractions. It is thus possible that our participants fatigued and did not produce maximum contractions at the end of the testing session. However, participants rested for one minute between each contraction which is long enough for strength recovery between trials [40] therefore the effect of fatigue on force production has been minimised. Finally, we assumed the relationship between the EMG amplitude and moment of the antagonist muscles to be linear, which may not always be the case particularly for untrained participants [71]. Additionally, our participants performed the maximal knee flexion and dorsiflexion contractions whilst seated and prone, respectively, and although changes of pressure on the electrodes were limited by securing the leg to the dynamometer, contact between the dynamometer and the electrodes may have modified the amplitude of the signal recorded. Though limited, this method allowed estimating the extensor moments of the knee and ankle and thereby provided a closer approximation of tendons force compared to using the net joint moment measured.

## 5. Conclusions

The present study showed for the first time that participants’ balance following exposure to a single slip with forward loss of balance does not rely on muscle strength, joint power, rate of moment development, electromechanical delay, or tendon stiffness of the leg extensors for this population of young (average twenty-five years) and older (average sixty-three years) participants. These novel findings further support the development of task-specific rather than exercise training fall prevention interventions [67].

## Figures and Tables

**Figure 1 sensors-22-03483-f001:**
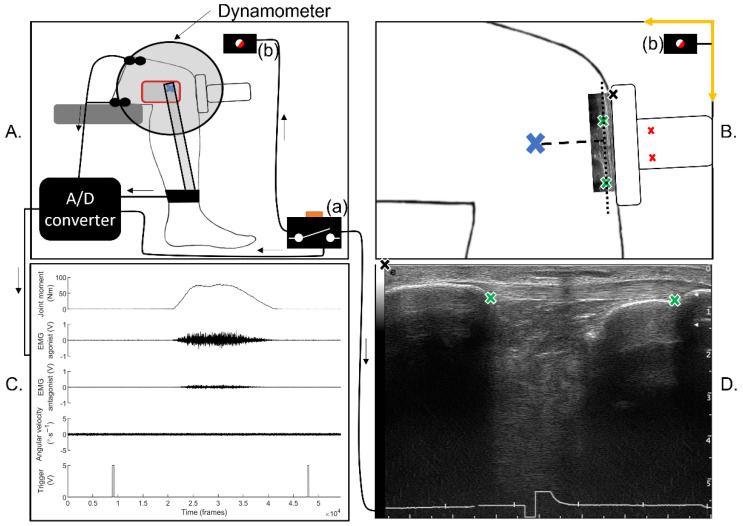
(**A**) Representation of the lab setup for the knee measurements. The red box represents the camera used to record videos, the lens was aligned with the joint centre represented by the blue cross. The trigger box (a) illuminated a LED (b) to sync the video and was captured by an analog to digital (A/D) converter as a 5 V signal and in the ECG input of the ultrasound video (shown on (**D**)). (**B**) Visual representation of the camera’s field of view, the anterior-posterior axis, vertical axis and coordinate system origin are represented by the yellow arrows, the LED light used to sync the videos is represented by (b), the blue cross represents the joint centre, the red crosses represent the markers placed on the ultrasound probe, the black cross represents the transducer origin (also represented on (**D**)), the green crosses represent the tendon’s proximal and distal insertions (also represented on (**D**)), the black dotted line represents the tendon’s line of action, the black dashed line represents the tendon’s moment arm. (**C**) Example of data collected from the dynamometer, EMG and trigger box. (**D**) Ultrasound image of the patellar tendon with the sync trigger visible at the bottom, black and green crosses represent the origin of the transducer and the tendon’s insertions, respectively.

**Figure 2 sensors-22-03483-f002:**
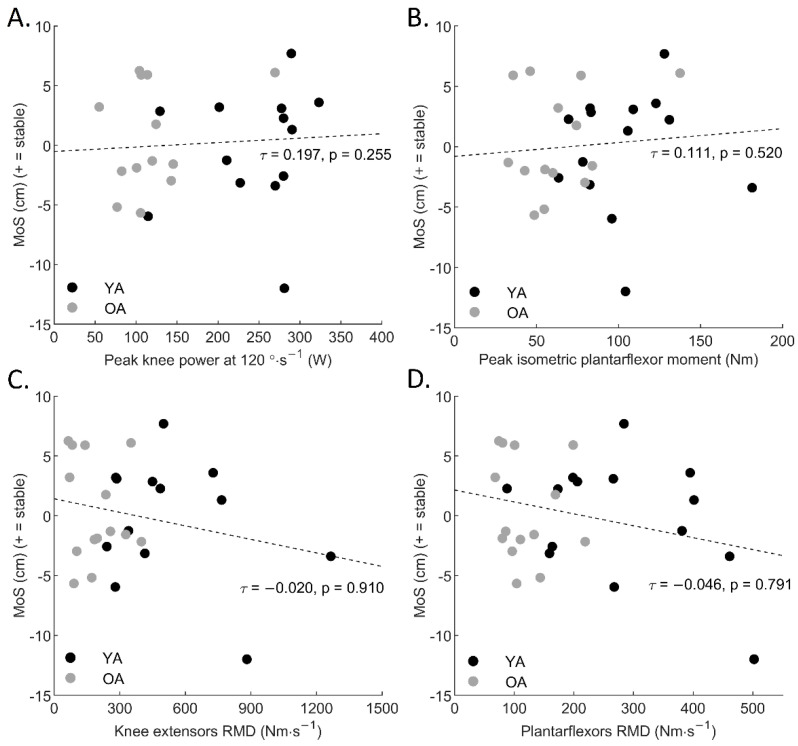
Correlations between (**A**) peak knee power at 120°·s^−1^ and margin of stability (*p* > 0.05), (**B**) peak plantarflexor moment and margin of stability (*p* > 0.05), (**C**) knee extensors rate of moment development and margin of stability (*p* > 0.05), (**D**) plantarflexors rate of moment development and margin of stability (*p* > 0.05). YA: young adults, OA: older adults.

**Table 1 sensors-22-03483-t001:** Summary of the tests performed to measure strength, RMD, EMD, power and tendon stiffness.

Joint	Outcome Measures		Contraction Type	Methods
Knee	Strength KE_MAX_		Isometric ramped 75°	Joint moment IKD
	RMD and EMD		Isometric maximum 90°	Joint moment IKD EMG activity agonist
	Power		Concentric 60° s^−1^ and 120° s^−1^	Joint moment IKD Joint angular velocity IKD
	PT stiffness	PT force	Isometric ramped 90° (M_KE_)	Joint moment IKD Ultrasound video 2D video
		PT elongation		
		MA_PT_		
		Antagonist moment	Isometric maximum knee flexion 90°	Joint moment IKD EMG activity antagonist
Ankle	Strength PF_MAX_		Isometric ramped 90°	Joint moment IKD
	RMD and EMD		Isometric maximum 90°	Joint moment IKD EMG activity agonist
	Power		Concentric 120° s^−1^	Joint moment IKD Joint angular velocity IKD
	AT stiffness	AT force	Isometric ramped 90° (M_PF_)	Joint moment IKD EMG activity antagonist
		AT elongation		Ultrasound video
		MA_AT_	Isometric ramped 90° (M_PF-MA_)	Joint moment IKD EMG activity antagonist Ultrasound video 2D video
		Antagonist moment	Isometric maximum dorsiflexion 90°	Joint moment IKD EMG activity antagonist

IKD: Isokinetic dynamometer, KE_MAX_: maximal knee extensor strength, RMD: rate of moment development, EMD: electromechanical delay, PT: patellar tendon, MA_PT_: moment arm of the patellar tendon, M_KE_: internal knee extensor moment, PF_MAX_: maximal plantarflexor strength, AT: Achilles’ tendon, MA_AT_: moment arm of the Achilles tendon, M_PF_: internal plantarflexors moment, M_PF-MA_: internal plantarflexors moment collected after M_PF_ with the ultrasound probe at 50% located at tendon length.

**Table 2 sensors-22-03483-t002:** Between group differences represented as medians (interquartile range) for young and older adults, and effect of age. Statistically significant *p* values in bold.

	Young Adults	Older Adults	Effect of Age
Knee extensor strength KE_MAX_ (Nm)	147.1 (94.5)	94.1 (49.2)	** *p* ** ** = 0.006**
Ankle plantarflexor strength PF_MAX_ (Nm)	100.1 (42.7)	57.5 (32.4)	** *p* ** ** = 0.001**
Knee power 60° s^−1^ (W)	158.9 (80.5)	73.4 (33.4)	** *p* ** ** < 0.001**
Knee power 120° s^−1^ (W)	277.4 (79.1)	106.2 (42.0)	** *p* ** ** < 0.001**
Ankle power 120° s^−1^ (W)	96.5 (45.1)	50.8 (16.5)	** *p* ** ** < 0.001**
Knee extensor RMD (Nm·s^−1^)	449.5 (462.7)	178.1 (186.6)	** *p* ** ** < 0.001**
Ankle plantarflexor RMD (Nm·s^−1^)	266.8 (225.8)	102.3 (69.6)	** *p* ** ** < 0.001**
Knee extensor EMD (ms)	23.1 (26.9)	24.1 (12.7)	*p* = 0.936
Ankle plantarflexor EMD (ms)	26.9 (13.0)	23.2 (8.4)	*p* = 0.616
Patellar tendon stiffness (N·mm^−1^)	549.6 (309.2)	449.2 (239.9)	*p* = 0.314
Achilles tendon stiffness (N·mm^−1^)	86.5 (38.4)	68.3 (41.6)	** *p* ** ** = 0.017**
Margin of stability (cm)	1.8 (6.3)	−1.5 (8.3)	*p* = 0.804

**Table 3 sensors-22-03483-t003:** Correlation table using Kendall’s tau between participants’ age or MTUs properties and balance.

	τ (Correlation with MoS)	BCa 95%CI	*p* Values
Age	−0.176	−0.531 to 0.239	*p* = 0.306
Knee extensor strength KE_MAX_ (Nm)	−0.072	−0.496 to 0.343	*p* = 0.677
Ankle plantarflexor strength PF_MAX_ (Nm)	0.111	−0.288 to 0.540	*p* = 0.520
Knee power 60° s^−1^ (W)	0.059	−0.339 to 0.447	*p* = 0.733
Knee power 120° s^−1^ (W)	0.197	−0.225 to 0.581	*p* = 0.255
Ankle power 120° s^−1^ (W)	0.059	−0.303 to 0.437	*p* = 0.733
Knee extensor RMD (Nm·s^−1^)	−0.020	−0.343 to 0.392	*p* = 0.910
Ankle plantarflexor RMD (Nm·s^−1^)	−0.046	−0.427 to 0.352	*p* = 0.791
Knee extensor EMD (ms)	0.066	−0.272 to 0.366	*p* = 0.705
Ankle plantarflexor EMD (ms)	−0.204	−0.549 to 0.125	*p* = 0.240
Patellar tendon stiffness (N·mm^−1^)	0.242	−0.288 to 0.673	*p* = 0.161
Achilles tendon stiffness (N·mm^−1^)	−0.020	−0.345 to 0.297	*p* = 0.910

## Data Availability

The data presented in this study are available on request from the corresponding author.

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
