# Peer review of "Role of Knee and Ankle Extensors’ Muscle-Tendon Properties in Dynamic Balance Recovery from a Simulated Slip"

_sensors, 2022, doi:10.3390/s22093483_

Round 1

Reviewer 1 Report

The study provides an insight into useful outcome measures as well as consequences regarding participants exposed to a simulated slip with forward loss of balance. The authors should be commended for this well written manuscript entitled, “Role of knee and ankle extensors’ muscle-tendon properties in dynamic balance recovery from a simulated slip” and for bringing this aspect of the study to light. Although the figures provided as supplementary material are clearly presented there seem to be no figure or diagrammatic illustration visually describing the experimental setup. Whilst this topic is considered interesting and timely there are a few concerns that need addressing.

Line 211-213: “The onset of EMG activation was set at 2 standard deviations above the average EMG activity recorded at rest over 250ms at the beginning of each trial before the contraction.” It is evident you could not report the SDs across the trials from all participants. Was there a reason for reporting 2 SDs? Was 2 SDs reflective of the SDs generated from the trials with respect to each participants’ output?

Line 262-266:A 5V external trigger was used to synchronise the ultrasound videos, camera videos and Acqknowledge data. This is very interesting and really should be displayed in a figure to appreciate the complexities involved in simultaneously collecting data using the above systems. Out of interest were the global / local coordinates aligned with the devices used?

Have many markers were placed on the ultrasound probe(s)? Three?

Also how many probes did you use for your measurement?

How did you maintain consistency whilst handling / placing the probe on the tendon? Was the Probe held in place by the experimenter’s hand during the data collection session or was the probe held in place using a third party holding device?

Reviewer 2 Report

The authors studied the influence of strength and MTU properties regarding the recovery of a slip forward loss in balance comparing young vs. old subjects. Result showed a loss in strength related parameters between subject groups, whereas statistics did not show a difference in tendon structures (Achilles+ patella tendon) and EMD. Further, now correlations were found between tested parameters and balance. A possible explanation for these results was the relatively young age of the old subjects and consequently the instability created by the protocol might be not challenging enough to differentiate between the age groups.

General:

The presented manuscripts adds some novel insight into the topic of falls (slip forward). The work is a nice example for the complexity of falls and shows the need for an ongoing research in this topic. In the conclusion, the authors should be more precise with respect to their study participants as a generalization of results is not be possible.

Specific comments:

Abstract:

L 13: Please include age of young and old subjects.

Introduction:

Overall, the introduction gives a good overview of the research topic and therefore provides a good basis for the study.

L 93: Tendons that are more compliant might not just change the force production capacity of the involved muscles based on the f-l-r but might also lead to a higher shortening speed of the fascicles and therefore a reduced force capacity because of the force-velocity relationship. This may lead to a change in the decoupling of tendon and fascicle shortening and hence influence the force-length-velocity properties of the involved muscle group. Please comment and add additional information in the introduction if necessary.

Methods:

L 126 - 127: The sample group in this study was heterogeneous with respect to the number of males and females in each age group. Do you think that this influenced the results? Are there results in the literature showing sex specific differences with respect to simulated slips?

L 131 – 132: Why was the study inclusion threshold set to 15 minutes unassisted walking? Is there literature supporting this criteria?

L 176: The authors mentioned appropriate familiarization trials. How did you check if the familiarization of the subjects was appropriate? Based on the reviewers experiences subjects often need several sessions to reproduce peak torque values or EMD values.

L177: What were the rest periods between trials? In L 130 the authors said that the MTU tests were done in a separate session. Was the order of the tests randomize and did you test for fatigue with e.g. an additional MVC at the end of this 2nd session?

L 211-214: Please add references for using 2SD a start threshold for EMG activity and joint moments.

L 218: Did you also correct for gravity during joint power assessment? What was the RoM for theses contractions?

L 227: Remove “dot” after s. “120°/s. “

L 219 – 220: Why did the authors measured the joint power at so many different velocities? Later you said that for better approximation of joint velocity during gait perturbation only 60°/s and 120°/s were chosen for further analysis. Why did you not include the results of your previous study before data collection of MTU properties? Hence, instead of 21 maximum dynamic contractions, participants should have made only 9 maximum dynamic contractions.

L 231-233: The authors said: “[…] during the isokinetic tests most older participants did not reach their maximal joint moment within the iso-velocity portion of the concentric contraction at 240°·s-1. Therefore, ankle joint power is reported at 120°·s-1 here.” Why did you not take the peak that occurred during the iso-velocity phase at 240°s-1 for power calculations?

L 235 – 239: Can you please explain this section more detailed for the reviewer?

L 251 – 255: You estimated antagonistic co-contractions assuming a linear relation between EMG and moment values. Because during this task subjects were sitting in the isokinetic device, was there any pressure on the EMG electrodes during the knee flexion contractions influencing the amplitude?

L 262: Out of curiosity: How did you send a 5V trigger signal to the iPhone?

L 284: Did you use different US devices because of the length of the probe? What was the sample rate of the devices?

Statistics:

L 322-323: You corrected the alpha level for the muscle-tendon properties to alpha = 0.005. The reviewer thinks this was due to multiple testing. However, is this necessary for your study design? Based on the reviewers understanding of the paper of Mark Rubin (2021, https://doi.org/10.1007/s11229-021-03276-4) you would not have to adjust the alpha level for the various MTU tests. Please comment.

Results:

The results section provides the necessary information for reader in an appropriate way.

Discussion:

L 361: Was the amount of lower strength and RMD comparable to previous studies?

L 366: Is the variability comparable to previous studies? Did you exclude outliers from analysis?

L 368: Is this large inter-individual variability in PT stiffness also present in previous published studies? If yes, do you think that future studies should include these measurements in their study design?

L 377: Beside tendon stiffness: Are there any other factors influencing EMD that might change with age?

L 377 – 380: This remains speculative and should be stated accordingly.

L 382 – 384: If the previous study did not show an effect on the recovery from a slip between ages, why did the authors expect that different PF and KE MTU properties are related to the balance of the same participants following a single slip with FLB as stated in L 116 – 118?

L 450 - 454: You mentioned several times that the relatively young age of your “old subject group” might be responsible for the missing of effects in your study. Therefore, your results might not be generalizable to “older” subjects in general.  Hence, please be more concrete in your conclusion of this study.

Round 2

Reviewer 2 Report

All questions and comments raised by the reviewer were adequately addressed.  The reviewer has no further comments. Nice work.